# The DNA Alkyltransferase Family of DNA Repair Proteins: Common Mechanisms, Diverse Functions

**DOI:** 10.3390/ijms25010463

**Published:** 2023-12-29

**Authors:** Ingrid Tessmer, Geoffrey P. Margison

**Affiliations:** 1Rudolf Virchow Center, University of Würzburg, Josef-Schneider-Strasse 2, 97080 Würzburg, Germany; 2School of Health Sciences, Faculty of Biology, Medicine and Health, University of Manchester, Manchester M13 9PL, UK; gmargison@manchester.ac.uk

**Keywords:** DNA repair, O6-alkylguanine-DNA alkyltransferase, alkylation damage

## Abstract

DNA alkyltransferase and alkyltransferase-like family proteins are responsible for the repair of highly mutagenic and cytotoxic O^6^-alkylguanine and O^4^-alkylthymine bases in DNA. Their mechanism involves binding to the damaged DNA and flipping the base out of the DNA helix into the active site pocket in the protein. Alkyltransferases then directly and irreversibly transfer the alkyl group from the base to the active site cysteine residue. In contrast, alkyltransferase-like proteins recruit nucleotide excision repair components for O^6^-alkylguanine elimination. One or more of these proteins are found in all kingdoms of life, and where this has been determined, their overall DNA repair mechanism is strictly conserved between organisms. Nevertheless, between species, subtle as well as more extensive differences that affect target lesion preferences and/or introduce additional protein functions have evolved. Examining these differences and their functional consequences is intricately entwined with understanding the details of their DNA repair mechanism(s) and their biological roles. In this review, we will present and discuss various aspects of the current status of knowledge on this intriguing protein family.

## 1. Introduction

DNA alkyltransferases are responsible for removing alkylation damage from DNA. As a background to exploring repair mechanisms dealing with alkylation damage in DNA, we first consider the generation and the mutagenic and cytotoxic potential of DNA alkyl lesions.

Alkylation of DNA bases and phosphodiesters in the DNA backbone can arise from both endogenous and exogenous agents. The former seems predominantly to be due to the nitrosation of amine-containing compounds and the subsequent metabolic activation of alkylating species by mixed function oxidases [1]. The methyl donor, S-adenosylmethionine, might also contribute to endogenous DNA methylation [1]. Exogenous agents include, for example, some constituents of cigarette smoke and certain cancer chemotherapeutic agents, although recent evidence suggests that there is a wide range of alkylation damage types in DNA and hence a large spectrum of environmental alkylating agents [2].

Alkylating agents can react with all the available N and O atoms in DNA [3,4], and the relative amounts of the alkylation products are determined by the mass and the chemical nature of the alkylating species, which react by either S_N_1 or S_N_2 type nucleophilic substitution (reviewed in [1,3,4,5]). The simplest examples of S_N_1 agents are MNNG (N-methyl-N’-nitro-N-nitrosoguanidine), the alkylnitrosoureas such as MNU (N-methyl-N-nitrosourea), nitrosamines such as NDMA (N,N-dimethylnitrosamine), and triazenes, including the cancer chemotherapeutic agents Temozolomide and Dacarbazine/DTIC. Examples of S_N_2 agents are MMS (methyl methanesulfonate) and DMS (dimethyl sulfate). In general, S_N_1 agents react more extensively with oxygen atoms, while S_N_2 agents attack mostly nitrogen atoms, but the relative amounts of the products are very different when comparing, for example, methylating and ethylating agents [6].

Several of the 12 DNA base alkylation products are mutagenic, clastogenic and/or cytotoxic, while alkylated phosphodiesters have not been reported to have significant adverse biological effects. It should be noted here that the biological effects of adducts have been established using predominantly methylating agents, and much less attention has been paid to the biological effects of higher alkylating agents. N^7^-methylguanines and N^3^-methyladenines are the most common products upon exposure to S_N_1 methylating agents (≤80% and ≤20% of the total reaction products, respectively) [4]. N^3^-methyladenine is mutagenic [7] and highly cytotoxic because it blocks the replicating polymerases [4,8,9] whereas N^7^-methylguanine is cytotoxic and mutagenic only after spontaneous depurination [4,5]. N^1^-methyladenine and N^3^-methylcytosine are also reported to be toxic because they are unable to base pair and consequently block DNA replication [10]. The most highly mutagenic alkylation products are the O^6^-alkylguanines and O^4^-alkylthymines. During replication, the former resulted in the mis-incorporation of thymine instead of cytosine, leading to G>A transition mutations, and the latter resulted in the mis-incorporation of guanine instead of adenine, leading to T>C transition mutations [11,12,13]. Furthermore, O^6^-alkylguanines are also highly cytotoxic because the post-replication mispairs trigger futile rounds of DNA mismatch repair, resulting in single-strand gaps, replication fork collapse, and double-strand break formation, which ultimately leads to cell death by apoptosis [14,15] or autophagy [16] (reviewed for example in [4]). This does not seem to be the case for O^4^-alkylthymines [17].

A plethora of DNA repair mechanisms that protect organisms against these adverse genotoxic effects have evolved, and these pathways and their key players have attracted attention over many decades. Base excision repair (BER), nucleotide excision repair (NER), mismatch repair (MMR), and three different types of direct damage reversal functions can all play a role in processing various types of alkylation damage. Alkylation damage repair processes are conserved in all organisms, and apparent redundancy within organisms is also common. The N^7^-alkylguanines and N^3^-alkyladenines, as well as some other N- and O-alkylation sites on DNA bases, are repaired by the BER system [1,5]. In the damage reversal pathways, in contrast to events in BER, NER, and MMR, the chemical modification is selectively targeted and removed from the damaged base without involving DNA backbone incisions. Two direct damage reversal systems exist in nature for the removal of alkyl lesions in DNA: the ALKB dioxygenases (ALKBH2 and 3 in humans) and the O^6^-alkylguanine-DNA alkyltransferases (AGTs, human version also known as O^6^-methylguanine-DNA methyltransferase MGMT). ALKBH proteins target alkyl groups at the N^1^ position of adenines and the N^3^ position of cytosines [18] (and reviewed recently in [1]). They act by oxidizing the alkyl groups in an alpha-ketoglutarate-Fe(II)-catalyzed reaction that results in the alkyl group being lost as formaldehyde. On the other hand, AGTs undertake the removal of alkyl groups attached to the O^6^ position of guanines and the O^4^ position of thymines in an autoinactivating (“suicide”) irreversible reaction that requires no cofactors and involves the transfer of the alkyl group to a cysteine residue in the active site pocket. An additional class of autoinactivating alkyltransferases (the methylphosphotriester (MPT) alkyltransferases) act on methylphosphotriesters in the DNA backbone. This MPT alkyltransferase function employs a different mechanistic strategy than used for the dealkylation of damaged guanines and thymines by AGT, although it also involves the inactivation of the protein by irreversible cysteine alkylation [19] (details in Section 2). Furthermore, within the AGT family are the structurally related alkyltransferase-like (ATL) proteins. These do not undertake damage reversal but flag the substrate O^6^-alkylguanine lesions for repair by NER [20,21] (details in Section 3 and Section 5).

DNA alkyltransferases are widespread in nature. Both primary amino acid sequences and crystal structures, and in some cases biochemical assays, have shown high levels of similarity at the functional, sequence, and structural levels among different proteins from this family.

The ability to remove potentially genotoxic lesions from DNA makes AGT an important player in cell survival, and by maintaining genome integrity, AGT protects cells against mutation and malignant transformation. In addition, human AGT has also become a focus of targeted inhibitor development because its repair activity counters the toxic effects of alkylation damage that is deliberately introduced into DNA by certain types of cancer chemotherapeutic agents [22,23,24].

In this review, we will discuss the adaptive response mediated by MPT alkyltransferase activity (Section 2), the repair mechanism in the AGT family of proteins (Section 3), the distribution of the different classes of alkyltransferases in nature (Section 4), interactions of these proteins with other protein systems and their functional implications (Section 5), and the inhibition and augmentation of AGT activity in cancer chemotherapy (Section 6).

## 2. The Adaptive Response: Where It All Began

In the 1970s, it was discovered that exposure of *E. coli* to low doses of the methylating agent MNNG increases resistance to a subsequent higher dose of this agent [25,26]. This phenomenon, not related to the SOS response in *E. coli*, became known as the adaptive response and is regulated by the *ada* gene. While O^6^-alkylguanine repair is mediated by the carboxy-terminal (C-terminal) part of the Ada protein [27], the adaptive response to DNA alkylation damage is triggered by the amino-terminal (N-terminal) domain of Ada repairing one of the stereoisomers of the methylated phosphotriesters (MPTs) in the DNA backbone. Subsequently, in the late 1980s, a second O^6^-methylguanine repair activity was identified in *E. coli* and the encoding gene, which has extensive sequence homology to the C-terminal domain of *ada*, was named *ogt* [28]. In contrast to Ada, however, the gene product OGT demonstrated exclusively O^6^-alkylguanine (and O^4^-alkylthymine) repair activity and was shown to be constitutively expressed, i.e., without alkylation dependent upregulation [28,29].

MPT methyltransferase activity in Ada is conferred via four conserved cysteine residues in two consensus motifs (CRPSC and PCKRC) in the N-terminal domain that coordinate a Zn^2+^ ion [19]. For Ada from *E. coli*, methylation of C38 in the first of these two motifs has been shown to trigger a conformational switch in the protein [19]. This conformational change is also referred to as an electrostatic switch since cysteine alkylation upon MPT repair reduces the negative charge in the Zn^2+^ coordinating region, which vastly enhances its DNA binding affinity [19]. This activates it as a transcription factor that binds to the ada box in the promotor region of the *ada* gene, upregulating its expression [30,31], as well as that of other ada box genes involved in the adaptive response to alkylation damage, i.e., AlkA, AlkB, and AidB [32,33]. This response is also triggered to a lesser extent by ethylating agents, but as far as has been reported, not higher alkylating agents, implying that it has evolved as a response to intermittent increases in the levels of, predominantly, methylating agents in the *E. coli* environment.

The adaptive response to alkylation has since been found in a number of other prokaryotes (see below, Section 4), but also in different *Aspergillus* species, which demonstrate MPT repair-mediated adaptive responses [34,35]. In contrast to *E. coli*, where O^6^-alkylguanine repair and MPT repair-coupled transcription upregulation are both located in the same Ada protein [33], in *Aspergillus*, these two functions are on separate proteins. [35]. A similar arrangement is found in the bacterium *B. subtilis*, which also contains two separate proteins for AGT and MPT alkyltransferase functions [35,36]. In addition to *Aspergillus*, *Vicia faba* has also been reported to show clastogenic adaptation by methylating agents [37], but the genes involved and the mechanism have yet to be defined.

Higher eukaryotes (including humans) do not manifest an *E. coli*-like adaptive response to DNA alkylation damage, probably because they do not express an MPT transferase activity. However, in animal models, AGT expression can be upregulated 2–4 fold in various tissues by pretreatment with alkylating agents, acetylaminofluorene, ionizing radiation, or, specifically for rat liver, partial hepatectomy [38]. The mechanisms for this are still unclear, although after ionizing radiation, at least in mice, the effect seems to be dependent on p53 expression [39]. It has also been suggested that p53 and protein kinase C (PKC) are involved in the modulation of AGT expression levels in tumors [40]. Although AGT is not itself converted by alkylation into a transcription factor for its own upregulation, it has been shown to modulate gene transcription through transcription factor interactions [41] (see Section 5).

## 3. Repair Mechanisms in the AGT Family

AGT family proteins are small proteins that typically consist of two domains (Figure 1A). Their mechanism of DNA repair is highly conserved and, as mentioned above, involves the irreversible transfer of the alkyl group from the O^6^ position of guanine or the O^4^ position of thymine onto a reactive cysteine in the protein. Crystal structures have demonstrated structural conservation between archaeal, bacterial, and eukaryotic AGTs (Figure 1B) [27,42,43,44,45,46,47] and, in particular, of several structural features that are directly linked to the highly conserved repair mechanism. Figure 1A shows these conserved elements, which are usually located in the C-terminal domain of AGTs: a nucleophilic cysteine for alkyl transfer from the damaged base, an active site loop, a helix-turn-helix (HTH) motif for DNA binding, an arginine finger that is important for base flipping, and an asparagine hinge. All AGT genes encode an active site pocket that contains a conserved I/V PCHR V/I V/I motif (see also Section 4 below), which harbors the nucleophilic cysteine (C145 in the human protein). Nucleophilic removal of the alkyl group from a guanine (or a thymine) base in the active site pocket is facilitated by interactions from neighboring amino acids (H146 and E172 in human AGT) that deprotonate and thus activate the cysteine (Figure 2) [48]. Subsequent transfer of the alkyl group to the cysteine moiety results in a restored guanine moiety and an alkylated AGT, which is then targeted for degradation (see below).

Although these structural features are found in all AGTs, subtle differences have evolved in the individual species that are intricately linked to varying abilities to bind to and repair different types of alkyl lesions. In the following sections, we examine the different functional elements of DNA alkyltransferases more closely.

### 3.1. DNA Interactions and Base Flipping by AGT

AGT binding to DNA is mediated by the conserved HTH motif in the C-terminal domain (Figure 1A and Figure 2A). In contrast to the usual HTH interactions, for example, in transcriptional repressors that bind sequence-specifically in the major groove of DNA, the HTH motif of AGTs binds in the minor groove with exclusively non-specific protein-DNA interactions [48], consistent with the observed lack of sequence specificity of AGT-DNA interactions [49]. In the DNA bound state, the highly conserved arginine finger (R128 in human AGT) intercalates into the DNA minor groove to facilitate flipping of the alkylated base out of the DNA double helix and into the active site pocket [48,50]. The arginine residue subsequently fills the space vacated by the flipped-out base [48], stabilizing the extrahelical conformation. Base flipping is further supported through steric interactions by a conserved tyrosine (Y114 in human AGT) [48]. Recent crystallographic studies of the thermophilic archaeal AGT from *Sulfurisphaera tokodaii* also suggested the role of the conserved tyrosine in protecting the active site cysteine from oxidation by blocking the binding pocket gate for access to oxidizing agents [45].

AGTs are also known to be able to dealkylate DNA bases in a single-stranded DNA (ssDNA) context. SsDNA binding is likely not based on interactions by the HTH motif itself but rather by the largely positively charged surface around the HTH domain [51]. For ssDNA, the damaged base does not need to be flipped out of a DNA duplex structure and can be directly bound in the active site pocket of AGT.

AGT also binds free guanine bases, and their nucleosides alkylated at the O^6^ position. In this case, binding occurs directly in the active site pocket of AGT (see below) and, in the relatively few cases where this has been determined experimentally, leads to the irreversible alkylation of the cysteine and inactivation of AGT. Free alkylated guanine substrates are the basis for inhibitors of AGT used in chemotherapy (see Section 6).

### 3.2. The Active Site Pocket

The active site pocket is formed by part of the DNA binding HTH motif, the asparagine hinge that links the active site and DNA binding motif, and part of the active site loop (see Figure 1A). The size of the substrate binding pocket varies between species, depending on the flexibilities of the pocket-lining elements and the presence of bulky amino acid residues (Figure 3). This determines the range of alkyl modifications that can be accommodated and repaired. For human AGT, the affinity and repair activity are stronger for some bulky alkyl lesions compared to small (methyl) modifications on guanine. Thus, repair rates follow the order: benzyl > methyl > ethyl > propyl/butyl [52,53] as a consequence of advantageous hydrophobic interactions, in particular by P140 within the substrate binding pocket (Figure 3A) [52,53]. A conserved lysine (K165) is also essential for O^6^-benzylguanine, and probably also other bulky O^6^-alkylguanines, processing by human AGT [54]. In contrast to human AGT, *E. coli* OGT shows no preference for O^6^-benzylguanines over O^6^-methylguanines, likely due to the reduced hydrophobicity of its binding pocket [55]. The alkyltransferase (AGT) activity of the *E. coli* Ada protein is even completely limited to smaller alkyl groups [47,55,56]. This has been attributed to a bulky amino acid residue (W160) at the upper part of the lesion binding pocket, blocking access for bulkier alkyl groups (Figure 3B) [47,55,56]. In addition, there is a partial loss of hydrophobicity of the pocket (P140 in human AGT is replaced by alanine in *E. coli* Ada) that is likely responsible for weaker interactions with the larger (hydrophobic) alkyl groups such as benzyl or isopropyl in the substrate lesions [56]. This probably also leads to the observed complete resistance of *E. coli* Ada to inhibition by O^6^-benzylguanine, which has also been reported for both Ada and OGT from *Salmonella typhimurium* [57]. Steric interference from bulky amino acid residues in the substrate binding pocket and reduced hydrophobicity of the pocket also cause a complete loss of affinity and repair activity for the bulky benzyl lesions by the AGT of the yeast species *S. cerevisiae* [58]. AGTs of other organisms, such as *Mycobacterium tuberculosis* or archaeal *Methanococcus jannaschii*, have evolved highly flexible substrate binding pockets, which likely enables them to accommodate larger alkyl lesions including benzyl [42,59,60]. Other archaeal AGTs (e.g., *Sulfolobus solfataricus*) can also readily transfer benzyl groups to the nucleophilic cysteine in their binding pockets [44]. For *M. tuberculosis* OGT, a repositioning of the active site loop towards the bound substrate has also been demonstrated [59], which may serve to stabilize ligand binding (Figure 3C). The high degree of flexibility in the substrate binding pocket may allow these proteins to optimize the fitting of the ligand binding cavity to the particular alkyl lesion type [59].

### 3.3. Catabolism of AGT Following Alkylation

The transfer of an alkyl group from a damaged base to the active site cysteine has been shown to destabilize the conformation of the protein (Figure 4A) [46] as well as to disrupt interactions between the C- and N-terminal domains [44,61]. In the thermophilic archaeon *Sulfolobus solfataricus,* destabilization results in a dramatically decreased melting temperature of 20 °C and 35 °C for the methylated and benzylated species, respectively [44]. Consequently, the protein “opens up”, as depicted in Figure 4B. This is consistent with the finding that alkylation of human AGT renders one of its lysine residues accessible to ubiquitination. While the primary target of ubiquitination is likely to be a single, specific lysine, this has not yet been unambiguously identified [62]. Ubiquitination then leads to the rapid degradation of AGT by the proteasome [62]. Ubiquitin-mediated proteasome degradation has been reported for humans, mice, yeast (*S. cerevisiae*), and archaeal AGT [62,63,64,65]. The extent to which this pathway occurs in other organisms has yet to be established.

### 3.4. The N-Terminal Domain

In comparison with the C-terminal domain, which contains the alkyltransferase and DNA binding activities of AGT, the N-terminal domain of AGT is not as conserved (Figure 1B). Available structures of bacterial OGT and Ada and some of the archaeal AGTs (e.g., *Pyrococcus kodakaraensis* and *Ferroplasma acidarmanus*) show an additional N-terminal helix in close proximity to the active site (compared to human AGT) [27,42,43,46,47] (S0ANZ2_FERAC AlphaFold database). This additional feature does not sterically restrict the insertion of the alkylated base into the active site pocket and, hence, does not interfere with alkyltransferase activity. However, in some organisms, for example, *Mycobacterium tuberculosis* and *Sulfolobus solfataricus*, it is involved in intra-domain interactions that have been suggested to play a role in the stabilization of the protein [42,44,61] as supported by the large drop in melting temperature upon destabilization of intra-domain interactions by AGT alkylation (as mentioned above) [44]. Crystal structures also revealed the tetrahedral coordination of a Zn^2+^ ion for human AGT (by C5, C24, H29, and H85) [46], and a disulfide bridge for archaeal AGTs (*Sulfolobus solfataricus* and *Sulfurisphaera tokodaii*, between residues C29 and C31) [44,45]. These different intra-domain interactions are believed to stabilize the N-terminal domain fold and, hence, overall protein stability. The N-terminal domain also stabilizes interactions with DNA and enhances alkyltransferase activity [42,44,61], possibly by capping the active site pocket that contains the inserted damaged base. However, the N-terminal domain of AGTs is not strictly required for alkyltransferase activity. In fact, some organisms have evolved fusion proteins that consist only of the C-terminal part of AGT fused to other protein functionalities, such as an endonuclease domain [66,67] or a domain with similarity to histones as in AGT2 from *Caenorhabditis elegans* [68,69].

Mutational analyses have also demonstrated that residues within the N-terminal domain are required for the formation of cooperative oligomeric complexes on DNA [70] that have been observed for human AGT (hAGT) by atomic force microscopy (AFM) imaging [49,51] and also in a crystal structure of *Mycobacterium tuberculosis* OGT [59]. Variants with modifications in the N-terminal domain of *M. tuberculosis* resulted in reduced DNA binding and significantly lower affinity for an alkyl lesion in DNA [42], suggesting that these cooperative interactions may play a role in stabilizing protein-DNA and protein-lesion interactions. These oligomeric interactions are considered in more detail in Section 5.

### 3.5. The Alkyltransferase-like (ATL) Proteins

Other members of the alkyltransferase family that do not possess an N-terminal domain are the alkyltransferase-like (ATL) proteins (Figure 5). ATLs share moderate sequence and high structural similarity with the C-terminal domain of AGTs (~30% sequence similarity between, e.g., human AGT and *E. coli* ATL [71])). In particular, all structural features for DNA binding and base flipping into the substrate binding pocket (HTH motif with arginine finger and tyrosine) are conserved between AGTs and ATLs (Figure 5A,B) [20,72]. Strikingly, however, these proteins do not possess the active site cysteine that is universal in AGTs: in most ATL proteins, the cysteine is replaced by a tryptophan, for example, in *E. coli* ATL or *S. pombe* Atl1 (Figure 5B). *E. coli* ATL, and presumably all other ATL proteins, have no in vitro alkyltransferase, glycosylase, or endonuclease repair activity [71,73]. Surprisingly, replacing the tryptophan in the active site sequence PWHRV with cysteine to generate the AGT sequence does not bestow alkyltransferase activity on *E. coli* ATL, so other essential residues are also absent or changed [71]. However, direct interactions between *S. pombe* Atl1 or *E. coli* ATL and the *E. coli* NER proteins UvrA [20,21] and UvrC [20] have been shown. In addition, studies using deletion strains of *S. pombe* lacking either Atl1, the NER endonuclease Rad13 (*S. pombe* homolog of XPG), or both also indicate interactions of Atl1 with the NER system for the repair of a range of O^6^-alkylguanines [20]. Binding affinities for O^6^-alkylguanines in DNA (in the form of short oligonucleotides) are high, in the low nanomolar to the sub-nanomolar range for ATL [20,73,74,75] compared to affinities in the low micromolar range for AGT [76,77,78]. Higher stabilities of lesion-bound complexes, in particular for very large alkyl groups that are poor substrates for AGTs [20,73,74,75], are mediated by a large substrate binding pocket in ATL [20] that is lined by a number of hydrophobic residues (Figure 5B) [74,75]. Their high binding affinities for O^6^-alkylguanines are consistent with a function to stably bind to and mark these lesions for NER.

In addition, the asparagine hinge that links the HTH motif and the active site in AGTs is missing in ATLs and turns into a capping loop that acts to stabilize substrate binding [72]. The lesion-ATL complex is further stabilized by a shift of the active site loop towards the flipped-out base in the lesion binding pocket, causing a switch from an open into a closed conformation upon lesion binding (Figure 5C), which locks the protein onto the DNA at the lesion [20,21,72]. Tighter binding of alkyl lesions by ATLs than by AGTs is also mediated by an additional C-terminal extension loop that contributes to DNA interactions together with the conserved HTH domain (Figure 5B) [20,72]. In addition, an extended N-terminal helix in ATL also contacts the DNA, and together with the C-terminal extension loop presses on the DNA in the lesion-bound complex, inducing significantly stronger DNA bending compared to AGTs (~45° instead of 15–30° by human AGT) [20,21,48,74,79]. Strong local DNA bending by ATL likely facilitates recognition by the NER system [20,21,74], which is known to target bulky and extensively DNA-distorting lesions. ATL-NER interactions will be discussed in more detail in Section 5.

## 4. The DNA Alkyltransferase Protein Family–Distribution in Nature

AGTs and their ATL homologs are highly conserved in nature, being found in all kingdoms of life, from bacteria to archaea and eukaryotes [20,38,65,71,72,73] (Figure 6). Some organisms possess only an AGT, for example, the budding yeast *Saccharomyces cerevisiae* and *H. sapiens;* others only an ATL, for example, the fission yeast *Schizosaccharomyces pombe* [73,80,81]. In some organisms such as *E. coli* [71,82], both AGT and ATL are present. Indeed, *E. coli* (and related prokaryotes) contains two different genes, *Ogt* and *Ada* that code for proteins with alkyltransferase activity. The Ada protein itself consists of two alkyltransferase domains, one acting on O^6^-alkylguanines and the other on MPTs, while in some other organisms, such as *Aspergillus*, the AGT and MPT functions are on separate proteins (see also above, Section 2).

As further examples of diversity, two different AGT genes have been discovered in the nematode *C. elegans* [68], one coding for a conventional form of AGT (AGT-1) and one coding for AGT-2, which showed distinct functions in meiosis and development [69]. The archaeal organism *Ferroplasma acidarmanus* encodes two different AGT variants, one of which is a fusion between an alkyltransferase and an endonuclease (see also Section 3) [66]. This fusion protein (AGTendoV) repairs O^6^-alkylguanine as well as other alkylation-induced DNA lesions that are usually targets of the BER pathway and some products of base deamination [66]. Furthermore, archaea have also been reported to contain a fusion of ATL and EndoV endonuclease [20]. It has been speculated that the EndoV domain in these fusion proteins may function in a manner similar to XPG, the eukaryotic NER endonuclease [83], that is recruited as part of the ATL-initiated NER pathway. Such direct fusion of ATL and NER functions in archaea would mimic the mechanistic link between ATL and NER (see Section 3 and below, Section 5).

While plants are less well studied [83,84], the removal of O^6^-methylguanine from DNA in the root tips of *Vicia faba* following treatment with MNU has been reported [37]. Removal was enhanced by clastogenic adaptation, as seen in *E. coli*, implying an Ada-like function. A potential MPT methyltransferase function may, therefore, exist in plants to protect their genome against the adverse effects of alkylation damage. While no peptide sequence homologs of AGT, Ada, OGT, or ATL have been found in higher plants, given the dissimilarity between the active region structures of AGT and PTMT, the possibility cannot be excluded that AGTs or MPTs with completely different active sites might exist in nature. However, an AGT gene homolog is present in unicellular algae, e.g., *Chlamydomonas Reinhardtii* (Figure 6).

**Figure 6 ijms-25-00463-f006:**
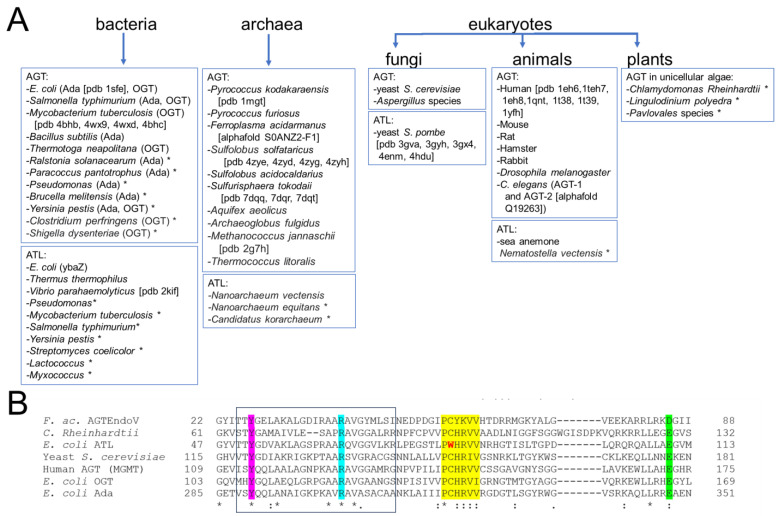
**Distribution of the DNA alkyltransferase superfamily.** (**A**) This overview shows currently identified species and is based on [20,34,35,36,38,42,43,45,60,61,65,66,68,71,72,75,85,86,87,88,89,90,91,92,93,94,95,96,97]. The pdb identifier for structural data is given in brackets where available. * indicates that alkyltransferase activity is only implied by sequence and has not been shown experimentally. (**B**) Exemplary multiple sequence alignment (using EMBL-EBI Clustal Omega) for the AGT/ATL protein variants from *E. coli*, OGT, Ada, and ATL, the human, yeast (*S. cerevisiae*), and *C. Rheinhardtii* AGT proteins, as well as the archaeal AGTEndoV fusion protein from *F. acidarmanus*. * indicates fully conserved residue, : indicates conservation of strongly similar properties of a residue. Different AGTs possess sequence identities of ~30–50% and similarities of ~40–70% (protein blast, NCBI). ATLs from different species have typical sequence identities of around 50% (e.g., *Vibrio parahaemolyticus* versus *E. coli*). Even between AGTs and ATLs, sequence conservation is high (e.g., 34% identity for *E. coli* ATL and human AGT). Highlighted are the highly conserved regions and residues: the consensus motif PCHRV/IV/I for AGT with C replaced by W in ATL in yellow; in pink the conserved tyrosine that supports base flipping; in cyan, the conserved arginine finger; in green the glutamate (or aspartate in AGTEndoV) that activates cysteine for alkyl group transfer as part of a catalytic triad together with histidine in the consensus motif (see Section 3). The HTH motif for DNA binding is boxed.

## 5. Functional Implications of Protein Interactions

Both AGT and ATL have been shown to form cooperative clusters on DNA at high protein concentrations (>4 μM; Figure 7 and Figure 8A,B) [21,49,51,70,98,99,100,101,102,103], while in the absence of DNA, they are predominantly monomeric [21,49,70,78,101] (Figure 7). It is worth noting that such cooperative clusters have yet to be demonstrated in a cellular context. Nevertheless, these data shed light on the probable lesion search and processing strategies of AGT and ATL, as outlined in the following sections. In addition to cooperative interactions between individual AGT or ATL monomers on DNA, both have been shown to interact with various other protein systems, and their role in repairing a large spectrum of different types of O^6^-alkylguanines and O^4^-alkylthymines has been investigated. Although far from complete, studies of these interactions have highlighted the potential complexity of the working mechanisms of these alkyltransferase family proteins, as discussed in the sections below.

### 5.1. Cooperative DNA Binding in DNA Lesion Search

In AGT, the protein-protein interactions for cooperative cluster formation, which are predominantly of an electrostatic nature, are located exclusively in the N-terminal domain (see model in Figure 8C) [51,70]. The model of the DNA-bound AGT clusters shows protein-protein interactions between each monomer with its third removed neighbor (e.g., AGT#1 interacts with AGT#4, AGT#2 interacts with AGT#5, etc. in the cluster via residues in their N-terminal domains). In the case of ATL proteins, which lack the N-terminal domain of AGT, interactions for cluster formation on DNA have been modeled to reside in completely different sites, i.e., in the very N-terminal helix and the C-terminal extension loop that is not present in AGTs (Figure 8D) [21]. These interactions are much weaker than those for AGT: there is a relatively small interaction interface in the crude structural model, and consequently, in contrast to AGT, DNA-bound ATL clusters require crosslinking to prevent their dissociation upon surface deposition in AFM imaging experiments [21].

Both AGT and ATL clusters on DNA have been shown to be limited in their lengths [21,49,101], with approximately four monomers per cluster suggested for ATL versus approximately seven monomers per cluster for AGT (from EMSAs and AUC studies using DNA of different lengths, as well as cluster lengths in AFM analyses) [20,21,48,49,101]. These maximum lengths may be restricted by the energetic cost from the bending strain induced in the DNA by each added monomer in the cluster, which eventually cancels out the energy gained from the cooperative protein-protein interactions [49]. The stronger DNA bending by ATL compared to AGT is consistent with shorter cluster lengths for ATL.

**Figure 8 ijms-25-00463-f008:**
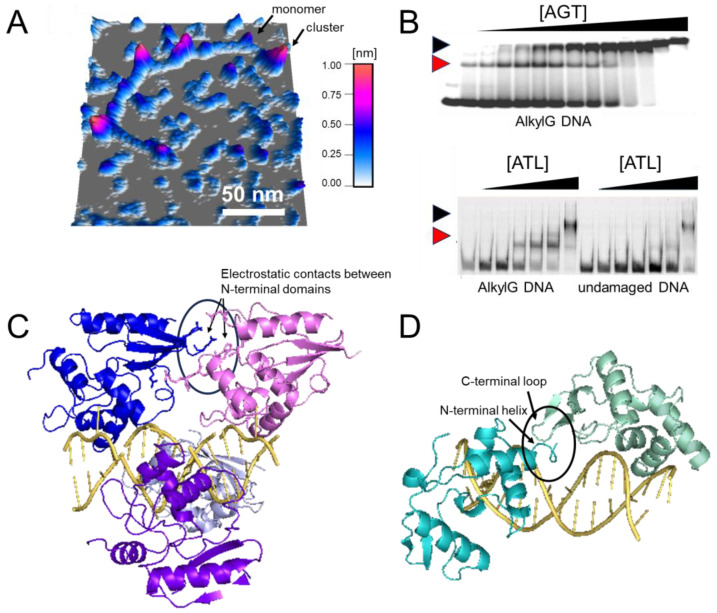
**Cluster formation on DNA.** (**A**) Atomic force microscopy (AFM) image of AGT clusters on undamaged DNA [49,104]. (**B**) Electrophoretic mobility shift assays of AGT (**top**) and ATL (**bottom**) [21,102]. Black and red arrows indicate clusters and lesion-specific complexes, respectively. Higher affinity for O^6^-methylguanine leads to the initial binding of monomeric AGT/ATL at the lesion, followed by cluster formation at higher protein concentrations (0–5.1 μM and 0–2 μM for AGT and ATL, respectively, with corresponding DNA concentrations of 150 nM and 50 nM). Note the one-step formation of clusters for the undamaged substrate indicative of cooperativity (shown only for ATL). (**C**,**D**) Models of AGT (**C**) and ATL (**D**) clusters on DNA [21,51]. AGT clusters are stabilized by protein-protein interactions between each monomer and its third removed neighbor (e.g., between blue and pink molecules at the top, as indicated by the oval). In ATL clusters, an N-terminal helix and the C-terminus extension loop form a weak interaction interface (as indicated by the oval). Figures in panel (**B**) have originally been published in *Nucleic Acids Research* and *PNAS* (modified from original), copyright at Oxford University Press and the National Academy of Sciences, respectively.

For cluster formation on ssDNA, the energetic cost from DNA duplex bending would be reduced, which would predict that clusters on ssDNA may be able to grow to much longer lengths than on dsDNA. Previous studies have shown, however, that the strength of cooperativity itself is also significantly lower for AGT on ssDNA compared to dsDNA [103], suggesting that cooperativity depends on the DNA duplex structure with minor groove binding functioning to orient the individual monomers. Studies with circular DNA with different degrees of supercoiling, as well as with G-quadruplex structures, show that cooperativity depends strongly on the correct juxtaposition of AGT monomers in the clusters, which appears to be optimized for relaxed, B-form DNA [103,105]. AGT monomer binding affinity for either duplex or ssDNA is not particularly high and only slightly (<10-fold) stronger for O^6^-methylguanine compared to undamaged DNA (in the low micromolar range, in either ssDNA or dsDNA context) [76,77,78]. The formation of AGT clusters on DNA may thus serve to stabilize and enhance DNA binding at high protein concentrations.

Preferential repair by AGT of O^6^-methylguanines towards the 3′ compared to the 5′ end of a short ssDNA substrate has suggested 5′-to-3′ directionality of AGT on DNA [48]. It has been speculated that cooperative clusters may play a role in enhancing the speed and efficiency of target site localization by preferential addition of monomer subunits at their 5′ ends and preferential dissociation from their 3′ ends [48,49]. However, recent studies using single-molecule fluorescence microscopy coupled with a dual trap optical tweezers system have demonstrated no enhancement of DNA translocation for AGT clusters versus monomers [101]. Furthermore, no directionality of AGT movement on DNA (either as monomers or clusters) has been observed in these single molecule visualizations of AGT cluster movement on DNA [101]. In fact, both AGT and ATL clusters moved bidirectionally on DNA, i.e., without any directional bias (Figure 9) [21,101]. At the same time, the ssDNA substrate in the studies that showed 5′-to-3′ bias in lesion repair had a length of only 70 nucleotides (nt), the single molecule fluorescence microscopy experiments employed long dsDNA tethers of almost 50,000 base pairs length. It, therefore, seems possible that directionality exists on ssDNA but that on dsDNA, AGT can switch between strands to achieve the observed bi-directional movement. However, repeating the experiments with ssDNA of the same (~50,000 nt) length [101] also showed movement without any directional bias. It is important to note that detection in single-molecule fluorescence microscopy studies is limited by pixel resolution to ~100 nm [101], which corresponds to almost 300 bp of DNA. It is possible that on the short length scale (≤70 nt corresponding to ≤~20 nm), 5′ to 3′ directional cluster growth led to the observed preferential repair of alkyl lesions located 3′ of the initial monomer binding site on DNA, while on the longer scale (>~300 bp or 100 nm) AGT clusters perform bidirectional, unbiased movement on DNA. Recently, a novel technology based on single molecule fluorescence quenching by graphene has enabled single-base pair resolution imaging of protein movement on DNA [106]. The method uses dsDNA pillars that are vertically attached to a graphene-coated surface and detects the movement of fluorescently labeled proteins on the DNA from changes in fluorescence lifetime due to distance-dependent fluorophore-graphene interactions. Using this method, bidirectional, single base pair steps by AGT monomers, as well as clusters, have been detected on DNA [106]. Future experiments using this novel methodology may allow the distinction between directional AGT cluster growth versus non-directional DNA scanning by the clusters.

DNA lesion search dynamics on DNA by AGT and ATL have been quantified using optical tweezers-coupled fluorescence microscopy and mean square displacement (MSD) analyses (Figure 9) [21,101]. AGT (monomers and clusters) shows a short-lived, fast-diffusing species and a longer-lived, more slowly moving species, which corresponds to diffusion constants that are lower than the theoretical limit for rotational diffusion along the DNA. These slowly diffusing complexes are thus consistent with the rotational movement of AGT tracking the minor groove of the DNA duplex. DNA translocation by ATL monomers, as well as clusters, appears to be faster than for AGT (Figure 9) [21,101]. This may be caused by the open conformation of ATL (Figure 5C) that prevails on undamaged DNA during lesion search in contrast to the less mobile, closed conformation of ATL induced by lesion recognition [21]. In addition, the translocation speed and pausing characteristics of ATL on DNA were not affected by interactions with the NER enzyme UvrA [21]. Together with the higher DNA binding affinity of ATL compared to AGT, this may thus enable ATL to rapidly transport NER components to ATL target lesions (see also below, section *AGT and ATL interactions with NER*). In this context, it has been speculated that additional DNA contacts by the EndoV domain in the archaeal ATLendoV fusion protein may enhance the speed of DNA translocation [83]. Future experiments might test this hypothesis.

### 5.2. Cooperative DNA Binding in DNA Lesion Processing

Recent single-molecule studies using a combined fluorescence-optical tweezers system also demonstrated preferential formation and/or stabilization of AGT clusters at an O^6^-methylguanine in DNA [101]. This is in contrast to the monomeric lesion-bound AGT complexes seen in crystal structures (e.g., Figure 3A) but is consistent with previous biochemical studies that also proposed a role of cooperative complex formation by AGT in lesion binding [102]. Analytical ultracentrifugation experiments showed a clear dependence of the oligomeric state of DNA-bound AGT on the protein:DNA ratio [101]; and while in crystallographic studies, the [protein]:[DNA] ratio is 1:1, single molecule fluorescence microscopy or biochemical assays typically employ [protein]>>[DNA]. Cluster formation may further stabilize AGT complexes on a target lesion, with the enhanced strain on the DNA from the additional monomer subunits in the cluster further enhancing the complete insertion of the alkylated base into the active site pocket of AGT. Furthermore, it might be speculated that the additional monomer subunits in the cluster would also be available for protein recruitment. For example, proteins from the replication machinery may be either held in place or newly recruited by the clusters to ensure rapid replication restart after replication stalling by stably bound AGT clusters at a lesion and subsequent repair of the alkyl base by AGT. Indeed, AGT has been proposed to interact with several proteins involved in DNA replication [107], as will be discussed in the next section.

Future experiments may also exploit methodologies such as single-molecule fluorescence imaging with high spatial and temporal resolution to study AGT and ATL interactions with different O^6^-alkylguanines and O^4^-alkylthymines in DNA (see *AGT and ATL interactions with NER*).

### 5.3. AGT Interactions with DNA Replication

Although AGT does not require any other protein factors for alkyl lesion repair, several interactions with proteins from other DNA repair and DNA processing pathways have been identified. Proteins in cancer cell extracts that were co-immunoprecipitated with AGT included the proliferating cell nuclear antigen (PCNA) clamp that serves as a platform for replication proteins on DNA and the MCM2 (minichromosome maintenance complex 2) component of the replicative helicase as well as the ORC1 origin recognition complex [107] (Figure 10). It should be noted, however, that these co-immunoprecipitation experiments were performed in the presence of DNA, so apparent interactions between proteins may in fact be due to mutual DNA binding. Direct physical contacts between AGT and DNA replication proteins hence remain to be demonstrated. Larger alkylguanines such as O^6^-pyridyloxobutylguanine (pobG) or O^6^-benzylguanine have been shown to present replication blocks [108], which can be overcome by specific translesion synthesis polymerases [109]. AGT binding to such lesions may further enhance DNA polymerase blocking and coordinate with the replication system to provide an important mechanism for the pre-replicative removal of mutagenic and toxic alkyl lesions.

### 5.4. AGT Interactions with DNA Mismatch Repair Proteins

Proteomic analyses have identified a potential interaction of AGT with MSH2 [107], which, together with MSH3 or MSH6, recognizes base-base or insertion-deletion mismatches in DNA to initiate the DNA mismatch repair (MMR) pathway. G:T mispairs, which arise from the misincorporation of thymine opposite O^6^-alkylguanine during DNA replication, are targets of the MMR machinery. As mentioned above, in the absence of alkyl lesion repair by AGT, futile rounds of MMR due to persisting alkylG:T mispairing in DNA replication eventually cause cell death [110]. Direct interactions of AGT with MSH2 may thus help to coordinate alkylation repair and MMR of alkylG:T mismatches during replication. Recruitment of AGT by MMR proteins might, by removing the alkyl lesion, terminate the futile and ultimately toxic MMR cycles. However, as for the interactions of AGT with DNA replication (see above), direct interactions between AGT and MMR proteins remain to be shown.

### 5.5. Roles of AGT in Transcription Regulation

Like the *E. coli* Ada protein [33], hAGT can also modulate transcription, although not of its own gene: upon alkylation, hAGT has been shown to directly interact with the estrogen receptor (ER) transcription factor [41]. This blocks ER activation by its coactivator and represses the production of cell growth-enhancing factors and cell proliferation [41]. DNA alkylation is thus translated by hAGT into a signal for cell cycle arrest, allowing more time for the synthesis of more AGT and, hence, to repair toxic (and mutagenic) alkyl lesions before the next round of replication. hAGT has also been shown to interact with the CPB/p300 histone acetylase [41], which modifies histones to open chromatin, allowing transcription but also making the DNA more vulnerable to alkylation (and other) damaging agents. ER targeted gene transcription modulation by hAGT may hence be fine-tuned by chromatin opening and ER inactivation. Future studies might reveal further interactions by AGT in transcription regulation.

### 5.6. AGT and ATL Interactions with NER

ATLs have been shown to directly interact with NER proteins [20,21]. In addition to an epistatic relationship with the eukaryotic NER endonuclease XPG, direct interactions with the prokaryotic NER initiating enzyme UvrA and the NER UvrC endonuclease have been demonstrated [20,21], supporting the direct recruitment of the NER system by ATL (Figure 11). Stronger binding affinity to larger alkyl lesions (e.g., O^6^-oxobutylguanine, pobG) lesions versus smaller lesions (e.g., O^6^-methylguanine) by ATL [20,74] (see Section 3) may play an important role in pathway selection for alkyl lesion repair. NER consists of two sub-pathways: global genome repair (GG-NER), which is independent of active transcription, and transcription-coupled repair (TC-NER), which is initiated by an RNA polymerase being stalled at a lesion. Weaker binding affinities for smaller alkyl lesions may, following recruitment, allow the NER proteins to displace ATL from the lesion for repair by the GG-NER sub-pathway, while larger alkyl lesions that have stronger binding affinities for ATL may lead to persistent ATL complexes on the DNA, which could stall RNA polymerase transcription and activate the TC-NER sub-pathway [74]. Clonogenic assays indeed demonstrated the involvement of the GG-NER sub-pathway in the ATL-associated repair of small alkyl lesions [74]. In contrast, cells containing bulkier alkyl lesions were more sensitive to the deletion of TC-NER-specific genes [74]. It is not known if stable clusters of ATL (rather than monomers) at a lesion, as have been observed in vitro for AGT at an O^6^-methylguanine (see above), might enhance RNA polymerase stalling and hence TC-NER initiation under certain conditions.

The co-localization of AGT with sites of active transcription [41,58,111] also hints at a potential role of AGT in the TC-NER pathway similar to that proposed for ATL on large alkyl lesions in DNA. Previous studies also implicated the NER pathway in the repair of O^6^-ethylguanine, O^6^-chloroethylguanine, and large branched-chain O^6^-alkylguanines [109,112,113,114]. Whether or not this is mediated by AGT binding to these lesions and whether the TC- or GG-NER pathway is involved has yet to be established.

Human AGT repairs methylated O^4^-thymine significantly slower and less efficiently compared to O^6^-methylguanine [115,116,117]. When expressed in a NER proficient *E. coli* strain but not in a NER-deficient strain, hAGT enhanced mutations arising from O^4^-methylthymines [117,118], suggesting that AGT may shield these lesions from repair by NER. The *E. coli* host cells in these experiments also contained an ATL, so a potential role of ATL in NER repair of O^4^-methylthymine cannot be excluded, although no binding of ATL to O^4^-alkylthymines and no role of ATL in their repair has so far been reported [119]. In addition, previous studies have concluded that O^4^-ethylthymine was repaired neither by the human AGT nor NER systems [112]. Ada proteins also showed no repair activity for O^4^-alkylthymines [120]. Hence, the mechanism of repair of the mutagenic O^4^-alkylthymines by AGTs and any role of NER still remains enigmatic.

### 5.7. Posttranslational Modifications of AGT

AGT has also recently been shown to directly interact with poly(ADP-ribose) polymerase 1 (PARP1) [121]. PARP1 functions in BER as well as other DNA repair pathways by adding poly-ADP-ribose chains (PAR) to its protein targets, as well as itself. These PAR chains can bind to several DNA repair proteins, leading to their enhanced recruitment to specific PARP1-marked DNA lesions [122,123,124]. For example, PARylation has been shown to be involved in the recruitment of the BER endonuclease APE1, the structure factor XRCC1 that plays a role in the organization of the BER mechanism, as well as other enzymes [124,125,126]. A recent study has shown that AGT directly interacts with PARP1 with affinities in the high nanomolar range (measured by microscale thermophoresis) and that AGT is PARylated by PARP1 (Figure 12A) [121], which enhances AGT activity in alkyl lesion repair [121]. In this context, the targets of AGT, O^6^-alkylguanines and O^4^-alkylthymines are not the only types of DNA alkylation damage (see for example Introduction). Hence, concomitant recruitment of AGT as well as BER factors by PARP1 may thus serve to ensure the rapid repair of clusters of different types of alkylation damages in DNA. The evolution of fusion proteins of AGT and the BER enzyme EndoV in archaea (see Section 3 and Section 4) may further support a potential coordination between the two pathways.

In addition to PARylation, other posttranslational modifications of AGT include ubiquitination, which, as mentioned above, destabilizes local AGT folding and targets the protein to the proteasome for degradation [62,127] and phosphorylation, which inhibits AGT repair activity (Figure 12B) [128,129,130]. Different kinases have been reported to potentially interact physically with AGT [130], and the level of phosphorylation correlated with the level of suppression of AGT repair activity [128]. It has been suggested that AGT deactivation by phosphorylation is due to shielding of the catalytic cysteine, possibly by phosphorylation of the highly conserved tyrosine 114 (Y114, Figure 12B) at the entrance to the active site pocket in AGT. Phosphorylation is hence suggested to provide protection of the reactive cysteine against modification in the cytosol. In addition, phosphorylation of serine 204 (S204) has been reported to enhance AGT resistance to proteomic degradation [38,131]. It is, therefore, reasonable to suggest that the phosphorylated form of AGT might prevail in the cytosol, while in the nucleus, alkaline phosphatases de-phosphorylate and thus activate AGT [128].

**Figure 12 ijms-25-00463-f012:**
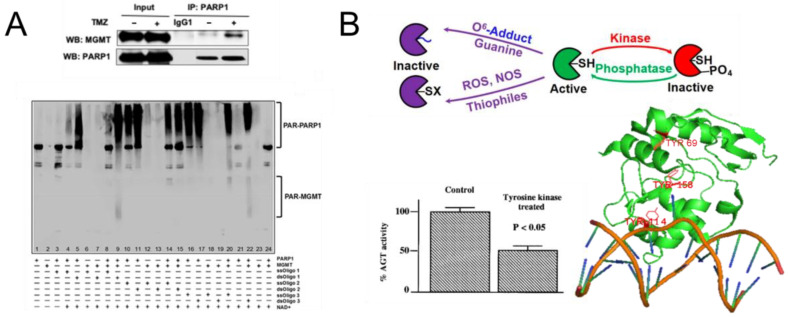
**Posttranslational modifications of AGT.** (**A**) Interactions between AGT and PARP1 were demonstrated by co-immunoprecipitation and western blot (WB) analysis after treating cells with temozolomide (TMZ) [121] (**top**). Immunoglobulin 1 (IgG1) served as a negative control. (**Bottom**): SDS PAGE/Western Blot (WB, PAR detection) showed the strongest PARylation of AGT by PARP1 in the presence of O^6^-methylguanine containing dsDNA (dsOligos1 and 3, lanes 9 and 22), as well as PARP1 autoPARylation [121]. (**B**) (**top**): Active AGT (green) is phosphorylated in vivo by kinases and dephosphorylated by phosphatases, and this reversible phosphorylation inactivates the protein (red). In addition, AGT is inactivated via reversible or permanent modification of the cysteine through reactive oxygen species (ROS), nitric oxide synthases (NOS), or thiophilic substances (purple), and by permanent alkylation through O^6^-alkylguanine repair (purple) [129]. The phosphorylation, in particular of tyrosines in AGT correlates with suppression of AGT repair activity (**bottom**) [130]. The structure of DNA lesion bound AGT highlights the positions of the three tyrosines in AGT (shown in red) that are candidates for functional modulation by phosphorylation (especially Y114, phosphorylation of which may block access to the active site cysteine and may interfere with DNA binding). Figures in panel (**A**) were originally published in *Journal of Hematology and Oncology* (modified from original), with copyright at Springer Nature Press. Figures in panel (**B**) have originally been published in *Cancers* (modified from original) and *Biochemical Journal*, copyright at MDPI and Portland Press/Biochemical Society, respectively.

## 6. AGT in Cancer Chemotherapy

Methylating agents (dacarbazine, temozolomide (TMZ), procarbazine, and streptozotocin) and chloroethylating agents (e.g., 1,3-bis(2-chloroethyl)-N-nitrosourea (BCNU)), among others, are used as cytotoxics in the treatment of various types of human cancers [132]. These drugs are also referred to collectively as O^6^-alkylating agents, and there is ample evidence that AGT provides protection against the toxic effects of these agents in cultured cells [133,134,135]. AGT expression levels have been determined for several human tumor types and also normal tissues and these have generally been based on functional assays of tissue extracts. A wide range of activities have been reported [38,136,137,138], and it seems reasonable to suggest that this may be the basis of the successful use of O^6^-alkylating agents only in certain tumor types. A number of single nucleotide polymorphic variants of AGT have been found, and while some of these have been correlated with protection against cancer induction [139,140,141,142], their possible contributions to the effects of O^6^-alkylating agent chemotherapy are so far not clear.

In the clinical setting, myelosuppression is a common dose-limiting toxicity, and this correlates with the low levels of expression of AGT in myeloid precursor cells [143]. In gliomblastoma multiforme (GBM) patients, methylation of the *agt* gene promoter correlates with better survival following chemoradiation involving Temozolomide [144], and promotor methylation is being used to stratify GBM treatment. AGT promotor methylation, which results in reduced or absent AGT expression as shown in cell culture models [135], has been reported in several other tumor types [145,146], but it is not yet known if these relate to tumor responses to O^6^-alkylating agent therapy [144,147].

The possibility that inhibition of AGT activity might be a strategy for enhancing the chemotherapeutic effectiveness in all tumor types treated with O^6^-alkylating agents has led to the synthesis and testing of a substantial number of candidate drugs. These “pseudosubstrates” are predominantly free-base guanines modified at the O^6^-position with a wide range of alkyl groups [148], although other compounds are also effective AGT inactivators [149]. Alkyl group transfer to the active pocket cysteine prevents the repair of O^6^-alkylguanine in DNA and, at least for O^6^-benzylguanine, results in AGT ubiquitination and degradation in the proteasome (see above) [62,150]. Human tumor xenografts grown in immune-deficient mice have been used to demonstrate the ability of these agents, principally O^6^-benzylguanine but also O^6^-bromothenylguanine (Lomeguatrib, LM), to enhance tumor growth inhibition by predominantly, TMZ or BCNU, and promising preclinical responses were obtained [151,152,153]. In cancer patients, after establishing that the dose of the agent required for AGT inactivation did not itself show any adverse side effects [154,155], phase I and II clinical trials have been carried out using inactivator-alkylating agents combinations [155,156,157]. However, the inactivators greatly exacerbated the off-target toxicities of the alkylating agents, initially requiring considerable dose reduction of the latter [155].

None of the clinical trials of these combination therapies have, so far, shown sufficient patient benefit to merit phase III trials for any tumor type or dosage regime for reasons that have yet to be established. Whether or not AGT inactivators that specifically target tumor cells [158,159,160] or other means of attenuating AGT activity, for example, tumor treating fields [161], alkylating drug combinations [162], or antisense strategies [163], will prove to be more successful remains to be seen.

No synthetic lethality has been reported for AGT. Given that MMR is required for O^6^-methylguanine to be lethal via futile repair cycles (see Section 5.4), loss of both AGT activity and MMR would not be expected to be synthetically lethal. Indeed, in AGT deficient cells, repeated treatment with the methylating agent N-methyl-N-nitrosourea has been shown to result in either reactivation of AGT expression or repression of the MMR pathway and increased survival [164,165]. In AGT-deficient or inhibited cells, inhibitors of other DNA repair pathways have been reported to be effective in increasing the in vitro toxicity of Temozolomide [166,167]. A compelling strategy would, therefore, appear to be to use cocktails of DNA repair inhibitors, of which a number are being trialed [168] in combination with O^6^-alkylating agents, and we look forward to future publications on such studies.

The myelosuppressive effects of O^6^-alkylating agents have been attributed to the low levels of expression of AGT in myeloid lineage precursor cells [139], suggesting the possibility that gene therapy using an AGT expression vector would resolve this problem. Mutant versions of AGT that are resistant to inactivation by pseudosubstrates but still able to repair O^6^-alkylguanine were shown to protect cells against the killing effects of these combinations [169,170]. Preclinical studies used retrovirus or lentivirus to deliver O^6^-benzylguanine or LM-resistant mutant AGT to cultured murine haemopoietic stem cells that were then reintroduced into myeloablated immune deficient mice and promising results were obtained using inactivator-O^6^-alkylating agent combinations [171,172]. In two clinical trials in glioblastoma patients, the P140K mutant of MGMT protected bone marrow against the toxic effects of TMZ and improved response rates. However, these studies involved 3 and 7 patients and it seems that no further trials have taken place over about the last decade [173,174].

As described elsewhere [175], O^6^-alkylating agents are toxic, and this is the basis of their use in chemotherapy, although the contribution that O^6^-alkylguanines make in eliciting cell senescence has yet to be explored [176]. Several products can be responsible, so focusing on AGT as the single resistance mechanism might be naive. But these agents are also highly mutagenic, so in addition to the treatments eliminating the more sensitive cells and selecting the more resistant ones, mutations in surviving cells might well engender novel resistance mechanisms, particularly when many treatment regimes involve repeat doses over long periods. At the same time, some of these agents are front-line therapies, so anything that can be done to improve the outcome is worth pursuing.

## 7. Concluding Remarks and Outlook

Members of the DNA alkyltransferase protein family are found in all organisms. They act on a variety of types of alkylation damage, specifically at the O^6^-position of guanine, the O^4^-position of thymine, and one of the phosphodiester oxygens in DNA. They probably evolved and are conserved to protect cells and organisms against the adverse biological effects of such damage, the origins of which are likely universal and may well be endogenous but remain largely unknown. Some members, the AGTs, remove the alkyl group from alkylated DNA bases, and that might require cooperative binding, although so far, this has only been shown in vitro. Others, the ATLs, do not themselves remove alkyl lesions but flag the damage for repair by NER or may have a NER component as part of their structure. These repair proteins that target alkylated DNA bases are highly conserved on a sequence and structural level and employ comparable DNA interaction mechanisms. On the other hand, the MPT transferase function that is required for the repair of methyl lesions in the DNA backbone and triggers the adaptive response involves different structural properties and operates by a different mechanism. The fact that the structures and mechanisms of action of AGTs and MPTs are so different might suggest that other family members may still have different mechanisms that have yet to be discovered. This might be the case in higher plants, at least one of which displays some characteristics of alkyltransferase function, but no sequence homologs.

While a substantial amount of detail has been revealed in the past decades on the repair of different alkylation lesions in DNA and on the structural and functional properties of AGTs and ATLs, unresolved questions remain. For instance, is there a role for AGT as an alkyl lesion damage-sensing protein in NER, analogous to ATLs? Can AGT clusters that have been shown to be established at O^6^-methylguanines in vitro also be detected in vivo? Are similar clusters formed on higher alkylated bases? And do they indeed play a role in protein recruitment to the lesion by AGT and ATL? What is the evolutionary, or perhaps environmental, significance of *E. coli* expressing three O^6^-alkylguanine processing proteins while other organisms (including humans) express only one, or perhaps none, in the case of higher plants?

In terms of practical applications, it is clear that alkylating agents are human carcinogens, and as long as alkylating agents are used in chemotherapy, a greater understanding of AGT has the potential to be highly beneficial in both the prevention and the treatment of human cancer.

## Figures and Tables

**Figure 1 ijms-25-00463-f001:**
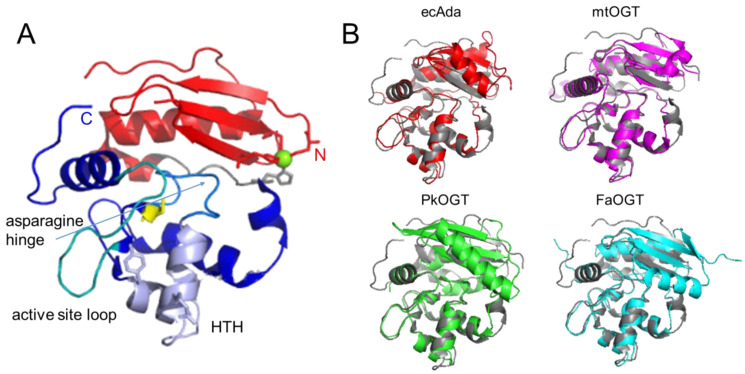
**Structural conservation of AGTs.** (**A**) Two-domain structure of human AGT (pdb 1eh6 [46]) with the C-terminal domain shown in blue colors, the N-terminal domain in red, and the connecting loop in grey. Conserved elements in the C-terminal domain are the active site cysteine in stick representation (yellow), the helix-turn-helix (HTH) motif (pale purple), including the conserved arginine (R128), and tyrosine (Y114, or phenylalanine in some species) in stick representation, the asparagine hinge that connects the HTH motif and the active site (light blue), and the active site loop (cyan). The Zn^2+^ ion coordinated by a tetrad of cysteine and histidine residues in the N-terminal domain is shown in green, and the coordinating amino acids in stick representation. (**B**) Crystal structures of the bacterial *E. coli* Ada (ecAda, red, C-terminal domain only, pdb 1sfe [27]) and *Mycobacterium tuberculosis* OGT (mtOGT, pink, pdb 4bhb [42]), as well as archaeal *Pyrococcus kodakaraensis* AGT (PkOGT, green, pdb 1mgt [43]), and the AlphaFold structural model of archaeal *Ferroplasma acidarmanus* (FaOGT, cyan, S0ANZ2-F1 AlphaFold Protein Structure Database), are shown overlaid with human AGT in grey.

**Figure 2 ijms-25-00463-f002:**
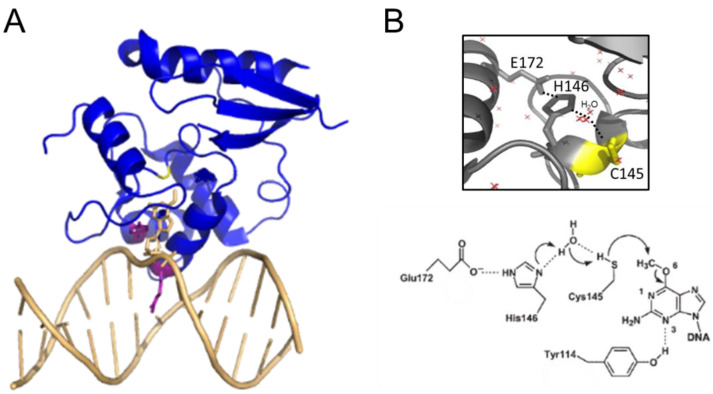
**AGT-DNA interactions.** (**A**) Structure of human AGT bound to an O^6^-methylguanine in DNA (pdb 1t38 [48]). The DNA is shown in light orange. The O^6^-methylguanine (stick representation) is shown flipped into the active site pocket of the protein, where it is attacked by the active site cysteine (yellow, replaced by serine in this variant to allow crystallization of a stable complex). The conserved arginine finger (R128) and tyrosine (Y114) that drive and stabilize base flipping are shown in purple in the stick representation. (**B**) Close-up view of the active site with the water-assisted hydrogen bonding (dotted lines) network between E172, H146, and C145 (yellow) that activates the cysteine nucleophile for de-alkylation of the damaged base (pdb 1eh6 [46]). Red crosses represent water molecules. A schematic of the dealkylation reaction is shown at the bottom (from [48]).

**Figure 3 ijms-25-00463-f003:**
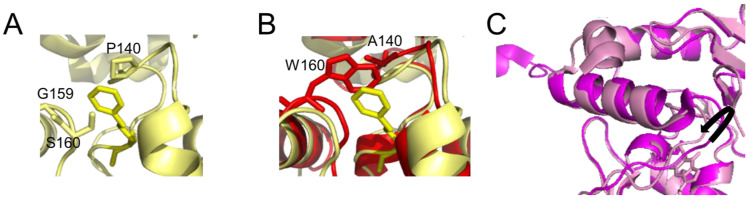
**Differences in the active site pocket.** Detailed view of (**A**) the active site pocket of human AGT with benzylated cysteine C145 (pdb 1eh8 [46]) and (**B**) *E. coli* Ada AGT (pdb 1sfe [27]) in red overlaid with the benzylated human protein in yellow. The benzyl group on C145 in the human protein is shown in yellow both in (**A**,**B**). In (**B**), in the *E. coli* Ada AGT, tryptophan (W160) would sterically clash with bulky alkyl groups (such as the benzyl) on the cysteine in its active site pocket. Furthermore, the hydrophobic P140 that interacts with the benzyl group in the human variant (**A**) is replaced by alanine (A140) in the (AGT) alkyltransferase active site of *E. coli* Ada (**B**). (**C**) Flexibility in the active site loop of *M. tuberculosis* OGT. The black arrow indicates the shift in the loop between the *apo* form of the protein (bright pink, pdb 4bhb [42]) and the protein bound to a lesion-mimicking chloroethyl analog (N^1^,O^6^-ethanoxanthosine) that covalently crosslinks AGT to the DNA [48,59] (pale pink, pdb 4wx9).

**Figure 4 ijms-25-00463-f004:**
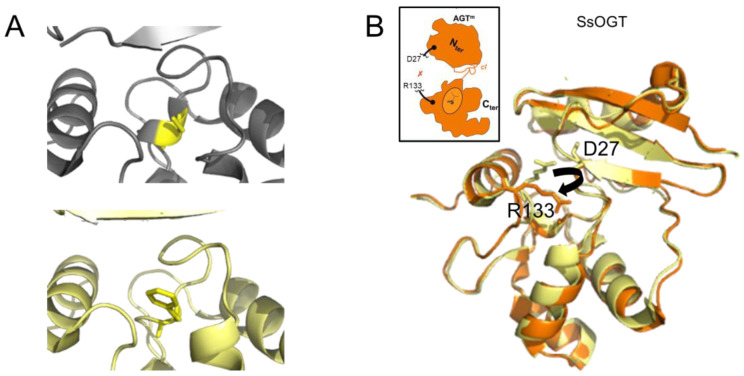
**AGT destabilization by alkylation.** (**A**) Accepting the benzyl group from O^6^-benzylguanine by C145 (yellow) results in the destabilization of a one-turn helix in the active site of human AGT (top: non-alkylated cysteine in a one-turn helix element, pdb 1eh6 [46]; bottom: benzylated cysteine within no secondary protein structure, pdb 1eh8 [46]). (**B**) In the thermostable archaeon *Sulfolobus solfataricus*, alkylation of the AGT active site cysteine leads to the disruption (black arrow in structure) of interactions between the N-terminal domain (D27 shown in stick representation) and the active site loop (R133, stick representation) that support structural stability at high temperatures. The non-methylated SsOGT is shown in yellow (pdb 4zye), and the methylated form is in orange (pdb 4zyg [44]). This results in an opening of the globular structure of the methylated protein, as indicated schematically in the inset, where the red x indicates the rupture of the D27-R133 interaction (schematic adapted from [44]).

**Figure 5 ijms-25-00463-f005:**
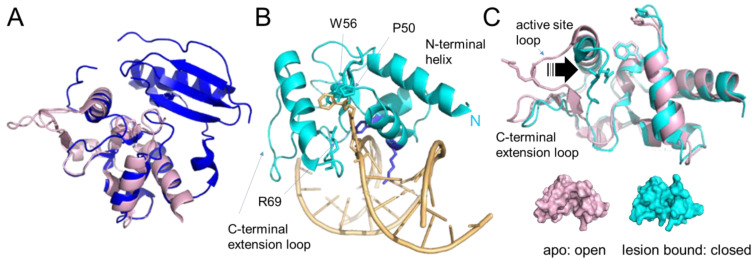
**Alkyltransferase-like proteins.** (**A**) Overlay of human AGT (blue, pdb 1eh6 [46]) and *S. pombe* ATL (Atl1, pdb 3gva [20], pale pink). Note the complete absence of the N-terminal domain in Atl1. (**B**) Atl1 bound to O^6^- benzylguanine in DNA (pdb 3gyh [20]). ATL is shown in cyan, the DNA in pale orange with the lesion base (in stick representation) flipped into the protein’s substrate binding pocket. The amino acids involved in interactions with the alkyl group (W56, P50) and with the flipped-out base (R69) are also shown in stick representation. The conserved arginine (R39) and tyrosine (Y25) that mediate base flipping by direct interactions with the DNA are shown in purple-blue. DNA contacts by the C-terminal extension loop and extended N-terminal helix enhance DNA bending. (**C**) An overlay of the unbound form of Atl1 from *S. pombe* (pale pink, pdb 3gva [20]) and the lesion-bound form (cyan, pdb 3gyh [20]) demonstrate the large shift of the active site loop (arrow) towards the binding pocket that allows R69 to interact with the damaged base. W56 (in place of the active site cysteine in AGT) and P50 of the binding pocket, as well as R69 on the active site loop, are shown in stick representation. The bound DNA and flipped-out alkylated base have been removed for clarity. Below, are surface representations of unbound and lesion-bound Atl1 to visualize the open-to-close conformational change in the protein upon lesion binding.

**Figure 7 ijms-25-00463-f007:**
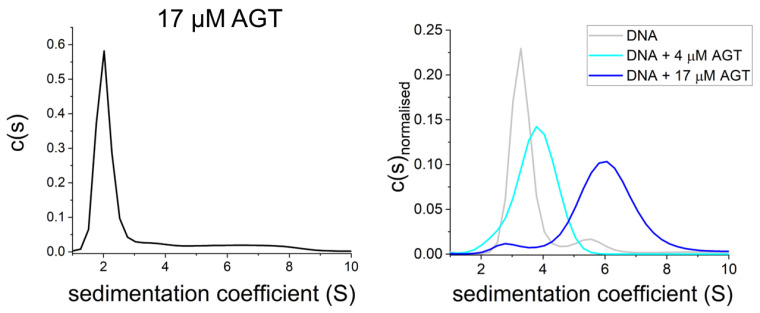
**AGT is monomeric in the absence of DNA.** Sedimentation velocity AUC shows predominantly monomeric AGT at high protein concentration in the absence of DNA ((**left**): 17 μM [78,101]) as well as on DNA at low micromolar concentration (4 μM, (**right**)), and oligomers of AGT on DNA at high protein concentration (17 μM, (**right**)) [101].

**Figure 9 ijms-25-00463-f009:**
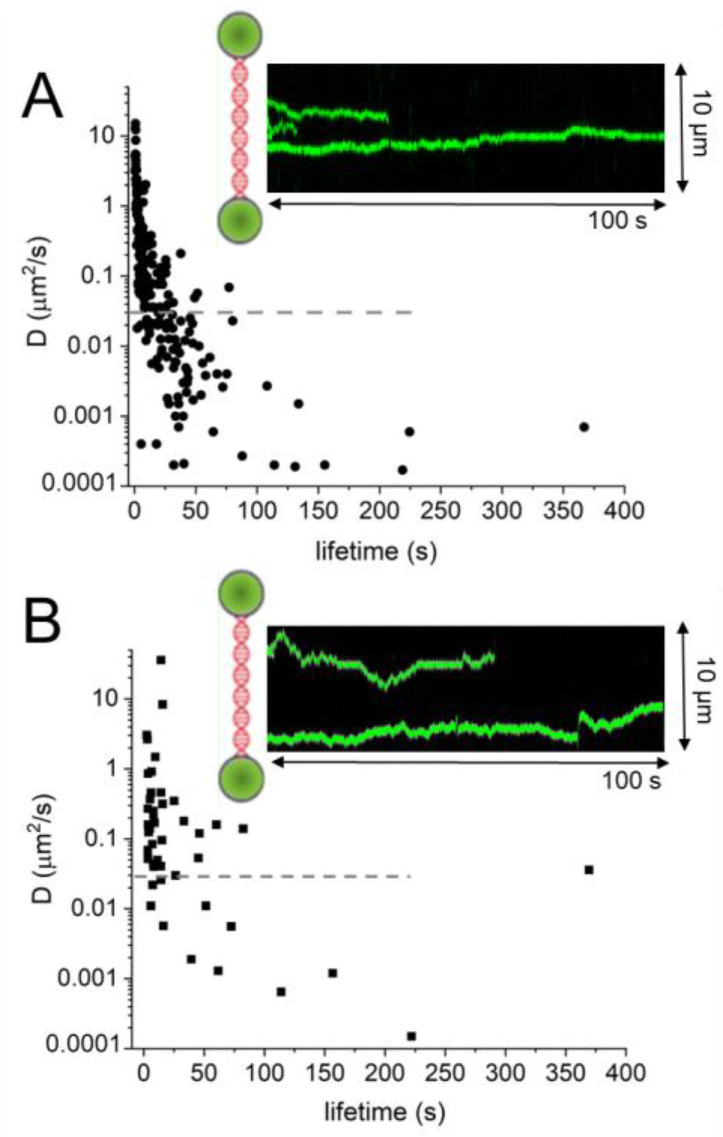
**DNA lesion search dynamics.** One-dimensional diffusion constants (D) on DNA plotted over the lifetimes of complexes on the DNA for (**A**) AGT [101] and (**B**) ATL [21]. The insets show representative kymographs (green traces) obtained by fluorescence microscopy-coupled dual trap optical tweezers, in which the y direction corresponds to the positions on the DNA tether (shown schematically between two beads held in the two optical traps), and the x direction to time. Mean square displacement analyses gave higher D values for ATL than for AGT: average values of 1.3 μm^2^/s for ATL versus 0.7 μm^2^/s for AGT. For AGT, higher diffusion constants predominantly stem from short-lived complexes (with lifetimes on the DNA of <10 s). The horizontal dashed lines in the D over lifetime plots indicate the theoretical limit for rotational diffusion of (quantum dot labelled) AGT and ATL along the DNA double helix. While these data were obtained at different protein concentrations (for ATL: 2 μM; for AGT: 4 μM) and in different buffers (for ATL: 25 mM HEPES pH 7.5, 25 mM Na-acetate, 10 mM Mg-acetate; for AGT: 10 mM Tris pH 7.7, 50 mM NaCl, 1 mM DTT), measurements with 4 μM ATL in the AGT experimental buffer gave comparable results as for the ATL buffer above (average D value of 1.5 μm^2^/s).

**Figure 10 ijms-25-00463-f010:**
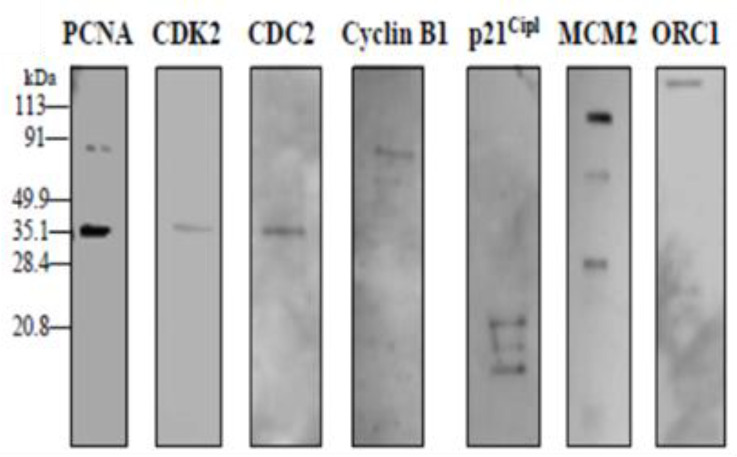
**Protein interactions by AGT.** Co-immunoprecipitation followed by proteomics identified several proteins from the replication machinery that potentially interact with AGT [107]. This work was originally published in *BBRC* (modified from original), and the copyright is at Elsevier.

**Figure 11 ijms-25-00463-f011:**
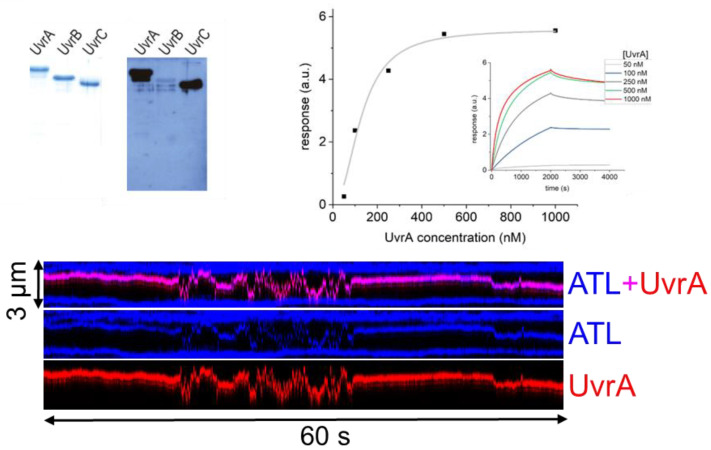
**Interactions between ATL and NER.** Direct interactions of *E. coli* ATL with the initiating enzyme of prokaryotic NER, UvrA, and the NER endonuclease UvrC, but not with the NER helicase UvrB were shown by immunoblotting for a FLAG-tag on ATL ((**top left**), modified from [20]). Parallel Coomassie staining on the left confirmed the presence of comparable amounts of UvrA-C in the gel. Biolayer interferometry (BLI, (**top right**) [21]) was performed with ATL immobilized on BLI sensors that were immersed in increasing concentrations of UvrA. Binding kinetics showed little dissociation over the time interval examined, and Hill fits (grey line) to the equilibrium binding signals (black squares) in the kinetic association-dissociation curves provided a K_D_ of ~100 nM for the interaction. Fluorescence kymographs from single molecule fluorescence optical tweezers studies (**bottom**) directly visualized the co-translocation of UvrA (red) and ATL (blue) on DNA, as seen in the overlay of the signals from blue and red detection channels (resulting in the pink trace [21]). The y axis gives the positions on the DNA (see also schematics in Figure 9), and the x axis is the time coordinate. This work has been originally published in *Nature* and *PNAS*, copyright at Springer Nature Press and the National Academy of Sciences.

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
