# Peer review of "The DNA Alkyltransferase Family of DNA Repair Proteins: Common Mechanisms, Diverse Functions"

_ijms, 2023, doi:10.3390/ijms25010463_

Round 1
Reviewer 1 Report
Comments and Suggestions for Authors
The review article is well drafted with highly relevant information about Alkyltransferase Family of DNA Repair Proteins.
1. A table representing a list of functionally characterized human AGTs along with their specific functions will be helpful.
2. Are any of the human AGTs druggable? What is the status of AGTs expression in human cancer tissues. What is the likelihood of AGT loss being synthetic lethal in cancers with certain genetic deficiencies such as microsatellite instability or BRCA deficiency. This should be discussed. The authors could check Cancer deepmap portal to obtain the relevant information.
Author Response
Response to Reviewer #1
- A table representing a list of functionally characterized human AGTs along with their specific functions will be helpful.
There is only one AGT in humans (referred to in our review as hAGT, also known as MGMT, O6-methylguanine methyltransferase, lines 79/80, line numbers in revised manuscript version) and a substantial amount of our review deals with its characterisation. To further clarify, we have now also added in section 6, line 847 (line number in revised manuscript, addition underlined): “What is the evolutionary, or perhaps environmental, significance of E.coli expressing three O6-alkylguanine processing proteins, while other organisms (including humans) express only one, or perhaps none in the case of higher plants?” However, there are a number of polymorphic variants and some of these have been partly characterised. To comply with the request, we now clarify these issues and include references to the AGT polymorphisms as follows (section 5 lines 752-755 in the revised manuscript):
“A number of single nucleotide polymorphic variants of AGT have been found and while some of these have been correlated with protection against cancer induction 139-142, their possible contributions to the effects of O6-alkylating agent chemotherapy is so far not clear.”
- Are any of the human AGTs druggable? What is the status of AGTs expression in human cancer tissues. What is the likelihood of AGT loss being synthetic lethal in cancers with certain genetic deficiencies such as microsatellite instability or BRCA deficiency. This should be discussed. The authors could check Cancer deepmap portal to obtain the relevant information.
Again, there is only one human AGT and we feel that, along with a brief mention in the introduction section, section 5 deals adequately with the clinical trials of drugs that inactivate the protein in the context of cancer chemotherapy. Therefore, we do not propose to make any changes with regards to this comment.
To comply with the question of AGT in cancer tissues, we now include references to these studies as follows (line numbering in revised manuscript):
Section 5 lines 748-752: “AGT expression levels have been determined for several human tumour types and also normal tissues and these have generally been based on functional assays of tissue extracts. A wide range of activities have been reported 38,136-138 and it seems reasonable to suggest that this may be the basis of the successful use of O6-alkylating agents only in certain tumour types.”
and lines 756-763: “In the clinical setting, myelosuppression is a common dose-limiting toxicity and this correlates with the low levels of expression of AGT in myeloid precursor cells 143. In gliomblastoma multiforme (GBM) patients, methylation of the agt gene promoter correlates with better survival following chemoradiation involving Temozolomide 144, and promotor methylation is being used to stratify GBM treatment. AGT promotor methylation, which results in reduced or absent AGT expression as shown in cell culture models 135, has been reported in several other tumour types 145,146, but it is not yet known if these relate to tumour responses to O6-alkylating agent therapy 144,147.“
To our knowledge there are no reports of synthetic lethality involving AGT. Nevertheless, to comply with the request to discuss this we have added (Section 5 lines 786-795, line numbers as in revised manuscript):
“No synthetic lethality has been reported for AGT. Given that MMR is required for O6-methylguanine to be lethal via futile repair cycles (see section 4.4), loss of both AGT activity and MMR would not be expected to be synthetically lethal. Indeed, in AGT deficient cells, repeated treatment with the methylating agent N-methyl-N-nitrosourea has been shown to result in either reactivation of AGT expression or repression of the MMR pathway and increased survival 164,165. In AGT deficient or inhibited cells, inhibitors of other DNA repair pathways have been reported to be effective in increasing the in vitro toxicity of Temozolomide 166,167. A compelling strategy would therefore appear to be to use cocktails of DNA repair inhibitors, of which a number are being trialled 168 in combination with O6-alkylating agents and we look forward to future publications on such studies.”

Reviewer 2 Report
Comments and Suggestions for Authors
The comprehensive review paper entitled "The DNA Alkyltransferase Family of DNA Repair Proteins: Common Mechanisms, Diverse Functions" authored by Tessmer and Margison provides a meticulous exploration of the DNA alkyltransferase family. The manuscript commences with a succinct introduction, followed by an elucidation of the historical milestones that delineate its inception. Subsequently, the review delves into the intricate repair mechanisms inherent in the AGT family, encompassing the C-terminal domain, active sites pocket, and N-terminal domain. The author systematically presents an in-depth analysis of the diverse DNA alkyltransferase protein families, thereby offering a nuanced understanding of their varied functions. Furthermore, the paper elucidates the pivotal roles these enzymes play in different cellular metabolism processes, contributing to a comprehensive comprehension of their functional significance.
Of notable significance is the author's discerning discussion on the potential implications of AGT in cancer treatment, underscoring the translational relevance of the DNA alkyltransferase family. Finally, the manuscript concludes by propounding avenues for future research, particularly in the realms of AGT and ATL, thereby guiding further exploration in this domain. In summation, the review manifests a methodical and well-organized approach, catering to the intellectual interests of readers in the International Journal of Molecular Sciences (IJMS).
Author Response
We thank all reviewers for their appreciation of our work.

Reviewer 3 Report
Comments and Suggestions for Authors
The review is well-written, informative, and well-organized. I have no comments and I recommend its acceptance in its present form.
Additional comment:
The authors present and discuss the DNA alkyltransferase family of proteins which are responsible for repairing DNA from alkylation damage. The authors examine different members of the family and their presence, conservation, and diversity in different kingdoms of life. The possibility of using inhibitors against these enzymes to assist in cancer treatment is discussed. The information provided would be useful to readers and help in understanding better DNA alkyltransferases and alkyltransferase-like proteins.Author Response
We thank all reviewers for their appreciation of our work.
